# Effects of Raster Angle and Material Components on Mechanical Properties of Polyether-Ether-Ketone/Calcium Silicate Scaffolds

**DOI:** 10.3390/polym13152547

**Published:** 2021-07-31

**Authors:** Jibao Zheng, Enchun Dong, Jianfeng Kang, Changning Sun, Chaozong Liu, Ling Wang, Dichen Li

**Affiliations:** 1State Key Laboratory for Manufacturing System Engineering, School of Mechanical Engineering, Xi’an Jiaotong University, Xi’an 710054, China; jibao_zheng@163.com (J.Z.); dongenchun@stu.xjtu.edu.cn (E.D.); sun.cn@xjtu.edu.cn (C.S.); 2Jihua Laboratory, Foshan 528200, China; kjfmg@sina.com; 3Institute of Orthopaedic & Musculoskeletal Science, Royal National Orthopaedic Hospital, University College London, Stanmore HA7 4LP, UK; chaozong.liu@ucl.ac.uk

**Keywords:** polyetheretherketone (PEEK), calcium silicate (CS), fused filament fabrication (FFF), mechanical properties, raster angle

## Abstract

Polyetheretherketone (PEEK) was widely used in the fabrication of bone substitutes for its excellent chemical resistance, thermal stability and mechanical properties that were similar to those of natural bone tissue. However, the biological inertness restricted the osseointegration with surrounding bone tissue. In this study, calcium silicate (CS) was introduced to improve the bioactivity of PEEK. The PEEK/CS composites scaffolds with CS contents in gradient were fabricated with different raster angles via fused filament fabrication (FFF). With the CS content ranging from 0 to 40% wt, the crystallinity degree (from 16% to 30%) and surface roughness (from 0.13 ± 0.04 to 0.48 ± 0.062 μm) of PEEK/CS scaffolds was enhanced. Mechanical testing showed that the compressive modulus of the PEEK/CS scaffolds could be tuned in the range of 23.3–541.5 MPa. Under the same printing raster angle, the compressive strength reached the maximum with CS content of 20% wt. The deformation process and failure modes could be adjusted by changing the raster angle. Furthermore, the mapping relationships among the modulus, strength, raster angle and CS content were derived, providing guidance for the selection of printing parameters and the control of mechanical properties.

## 1. Introduction

Porous scaffold has been used in tissue engineering to provide an appropriate environment and architecture for the tissue regeneration and mechanical properties control [1,2,3]. Compared with conventional manufacturing techniques, additive manufacturing (AM) techniques exhibited attractive attention for the abilities of constructing complex geometries [4], such as topology optimization structure and bionic bone trabeculae structure [5]. AM technologies mainly included stereolithography (STL) of a photosensitive resin, fused filament fabrication (FFF) of polymer and selective laser sintering (SLS) of metal powder [6]. Specially, the FFF technique was most widely used for its low cost of desktop printers and good manufacturing freedom [7]. Meanwhile, the FFF printing parameters, such as nozzle temperature, line width and raster angle, could be adjusted as required.

In recent years, polylactic acid (PLA) [8], polycaprolactone (PCL) [9] and polyetheretherketone (PEEK) [10] were used to fabricate the porous scaffold by the FFF technique. PEEK, as a semi-crystalline polymeric material, exhibited excellent chemical resistance and thermal stability [11,12]. Moreover, the mechanical properties of PEEK (such as strength and Young’s modulus) were similar to those of natural bone tissue [13], which reduced the stress shielding effect and improved the load transmission [14]. Thus, PEEK was widely used in clinical applications, such as interbody fusion cage [15], rib [16], skull [17] and mandible [18]. Meanwhile, PEEK was expected to replace metal materials such as titanium alloys and stainless steel in the non-load-bearing environment. However, the biological inertness and poor osteogenic capability restricted the clinical application.

To improve the bioactivity of PEEK, many studies were committed to fabricate PEEK-based composites via the incorporation of additives such as hydroxyapatite (HA) [19,20], calcium phosphate (TCP) [21], bioglass (BGA) [22] and calcium silicate (CS) [23,24,25]. Specially, CS was demonstrated to present excellent bioactivity and the apatite formation on CS was faster than that of BGA [26]. Numerous studies found that CS could promote the proliferation, adhesion and differentiation of osteoblasts and bone-marrow mesenchymal stem cells [27,28,29]. Furthermore, a higher bone contact ratio and new bone formation was found for PEEK/CS than that of PEEK/HA in animal models. The above studies [28,29,30] demonstrated that the participant of CS could effectively improve the osseointegration ability of PEEK. However, the studies in FFF printing of PEEK/CS composite scaffolds were rare because of the complicated fabrication procedure.

In addition to the bioactivity, the mechanical properties of the scaffolds could affect the load transmitting mechanisms and tissue growth. For the composites fabricated by FFF, the printing parameters and material component could impact the mechanical performances [31,32]. Chacón et al. [33] investigated the effects of printing orientation on the mechanical properties of PLA and found that specimens showed the optimal mechanical performance in terms of strength, stiffness and ductility along the 0° raster angle. Zhang et al. [34] studied the mechanical properties of PLA and aluminum fiber-reinforced PLA composites, who found that the addition of aluminum fiber improves the dynamic mechanical thermal property, while the tensile strength and elastic modulus decreased. Thus, the printing orientation and material composition had a major impact on the mechanical properties [35,36]. The above-mentioned research mainly focused on solid printed specimens; however, limited studies were found in the literature on the mechanical properties of PEEK-based composites scaffold with different printing orientation and CS content.

In this work, PEEK/CS composites scaffolds with different CS contents and raster angles were printed by FFF 3D printing technology. The surface micro-topography, surface roughness and internal structure of PEEK/CS scaffolds were investigated. The crystalline and thermal behaviors of the PEEK/CS composites were also determined. The influences of the CS content and raster angle on the mechanical properties of the PEEK/CS scaffolds was systematically studied. Moreover, the deformation process of the composite scaffolds was recorded using a high-speed camera to analyze the deformation mechanism.

## 2. Materials and Methods

### 2.1. Fabrication of PEEK/CS Composite Scaffolds

Figure 1a exhibits the fabrication process of the scaffolds based on FFF 3D printing. PEEK and CS powder were mixed with mass ratios of 8:2 and 6:4. The PEEK/CS composites were fed into a twin-screw extruder and extruded into filaments with diameter of 1.75 mm. The scaffold was fabricated using Engineer 200 from JuGao Technologies. Cubic scaffold specimens (length, width and height of 10 mm) were printed by FFF process. To evaluate the effect of printing direction on the mechanical properties, the raster angle was set as ±15°, ±30°, ±45°, ±60°, ±75° and ±90°. Figure 1b shows schematic diagram of the scaffolds with different raster angles. The rest of the printing parameters were set as follows: nozzle temperature of 420 °C, layer thickness of 0.2 mm, line width of 0.4 mm, line spacing 0.4 mm and printing speed of 30 mm/s. The scaffolds were named after the CS content raster angle. For example, 20%CS-15° represented that the CS content was 20 wt.% and the raster angle was ±15°.

### 2.2. Methodology for the Characterizations of the Composite Scaffolds

Micro-computed tomography (Micro-CT, Y.Cheetah, YXLON, Hamburg, Germany) was used to measure the internal structures of the pure PEEK and PEEK/CS composite scaffolds. The surface morphologies of the scaffolds were characterized via scanning electron microscopy (SEM, su-8010, Hitachi, Japan) after being coated with Au. The acceleration voltage and electric current was set as 10 kV and 20 μA, respectively. Laser scanning confocal microscopy (LSCM, OLS4000, Olympus, Japan) was used to evaluate the surface roughness (Ra). The porosity of scaffolds with different CS content was further calculated based on weighting method.

### 2.3. Thermal Behaviors of the Composites

Fourier transform infrared spectrometry (FTIR, Nicolet iS10, Thermo Fisher, Waltham, MA, USA) was used to measure changes of chemical bonds at room temperature. The spectra of the PEEK/CS composite were measured in the frequency range of 0–3000 cm^−1^. The thermal behaviors of pure PEEK and PEEK/CS were determined by differential scanning calorimetry (DSC, DSC1, Mettler Toledo, Columbus, OH, USA). Samples of around 3–5 mg in weight were heated from 20 to 400 °C at 10 °C/min (heating rate). According to the DSC results, the glass transition temperature (*T_g_*) and melting temperature (*T_m_*) for PEEK and PEEK/CS composites were revealed. The crystallinity degree of the composites was calculated by the formula expressed in Equation (1):(1)$$X_{c}=\frac{\Delta H_{m}-\Delta H_{c}}{\Delta H_{f}\times W_{PEEK}}$$
where $X_{c}$, $\Delta H_{m}$ and $\Delta H_{c}$ represented the crystallinity, melting enthalpy and crystal enthalpy of composites, respectively. $\Delta H_{f}$ is the enthalpy of fusion of a 100% crystalline PEEK sample (130 J/g). $W_{PEEK}$ is the mass fraction of PEEK in the PEEK/CS composite.

### 2.4. Mechanical Testing

To assess the influence of CS content and raster angle on the mechanical properties of scaffolds, PEEK/CS scaffolds were compressed by a mechanical testing machine (CMT4304, MTS Corp, Eden Prairie, MN, USA) at a compression speed of 3 mm/min. The stress–strain curves were obtained from the load-displacement data to calculate the Young’s modulus and the strength. Meanwhile, the compressive responses of the PEEK/CS scaffolds were recorded using a high-speed camera and the deformation mechanism was analyzed.

## 3. Results and Discussion

### 3.1. Micro-Structure Characteristics

Micro-CT was used to evaluate the interconnectivity of the scaffolds, as shown in Figure 2a. No obvious defect on the 3D and top views was found, indicating that the FFF process was suitable for the manufacturing of PEEK/CS composites. Meanwhile, the pure PEEK and PEEK/CS composite scaffolds exhibited good interconnectivity, shown as the top view. The SEM results (Figure 2b) found that the pure PEEK scaffolds exhibited a smooth surface, while a coarser topography was found on the surface of PEEK/CS scaffolds. Furthermore, the LSCM was used to quantitatively measure the surface roughness. The 0.13 ± 0.04 μm was found for the pure PEEK scaffold, while the 20%CS and 40%CS scaffolds exhibited the *Ra* of 0.38 ± 0.09 and 0.48 ± 0.06 μm, respectively. The LSCM results (Figure 2c) demonstrated that the participation of CS particle could improve the surface roughness of scaffolds, which may be beneficial to cell attachment and spread [37,38]. Table 1 shows that pure PEEK and 20%CS scaffolds exhibited similar porosity (around 55%), while the porosity of the 40%CS scaffolds increased to 62%. This may be caused by the addition of CS particles, which improved the viscosity of PEEK material in the molten state, consequently causing the decrease of output of the nozzle. The above phenomenon finally resulted in the decrease of the porosity. In addition to printing parameters, the consistency of the scaffolds structure could be impacted by CS content. The different printing parameters should be studied to compensate for the impact of CS content variation in the future.

### 3.2. Thermal Behavior of Composites

Figure 3c,d exhibit DSC cures and crystalline degree results. The peak around 172 °C and 345 °C represented the glass transition and melting process for the pure PEEK. As the CS particles act as a nucleating agent and reduce the nucleation barrier of crystallization, there was no evident glass transition peak for the PEEK/CS composites. The crystalline degree increased from 16% to 30% with the CS content ranging from 0 to 40%wt (Figure 3d). Generally, excessive additives would hinder the crystallization of the polymer. In this study, the CS volume fractions of the PEEK/CS scaffolds with CS contents of the 20, and 40 wt.% were 11.2 and 23.1 Vol.%, respectively. The CS content of 40 wt.% did not reach the critical value that hindered the movement of PEEK molecular chains, which resulted in the continuous increase of crystalline degree. Previous studies found that the crystallinity of PEEK could be adjusted by controlling the ambient temperature of the printing process. The results in this study indicated that the additive contents also exhibited an important impact on the crystallinity of PEEK.

The XRD pattern of pure PEEK and PEEK/CS is showed in Figure 3a. A diffraction peak of PEEK was found at 18.7°. Owing to the increase of crystalline degree, three obvious diffraction peaks of PEEK were observed at 18.7°, 20.7° and 22.6° for the PEEK/CS composites, which verified the DSC results. The diffraction peaks of CS were mainly distributed at 25.4°, 26.8°, 28.8° and 29.9°. As CS particles cover the PEEK matrix, the diffraction peaks of CS enhanced with increase of CS content. Figure 3b showed the FTIR result and the characteristic peak at 1596 and 1184 cm^−1^ represented the C=C and C–O–C of PEEK, respectively. The characteristic peak around 1043 and 861 cm^−1^ indicated the vibration of Si-O-Ca and Si-O-Si for the PEEK/CS composites. The XRD and FTIR results demonstrated uniform distribution of CS particles on the composite surface. Meanwhile, no new characteristic peaks were found in the XRD and FTIR results, indicating that the addition of CS particles did not affect the thermal stability of PEEK/CS composites.

### 3.3. Mechanical Properties

The compressive modulus is shown in the Figure 4a. The Young’s modulus of three types of scaffolds remarkably increased with the raster angle ranging from 15° to 90°. The modulus variation ranges of pure PEEK, 20%CS and 40%CS scaffolds were 28.8–406.9 MPa, 63.2–485.1 MPa and 23.3–541.5 MPa, respectively. Under the same printing raster angle, the modulus of 20%CS scaffolds was higher than that of pure PEEK and 40%CS scaffolds when the raster angle was less than 90°. Owing to the excellent hardness and stiffness of biological ceramic, the addition of CS particles generally enhanced the modulus of polymer-based composites, while the modulus of 40%CS scaffolds was lower than that of 20%CS scaffolds in this study. This phenomenon may be caused by the lower porosity of 40%CS scaffolds, as shown as Table 1. The decrease of porosity of 40%CS scaffolds could reduce the rod diameter of scaffolds and the effective bearing area under the compressive load, finally leading to the decrease of Young’s modulus. Under the raster angle of 90°, the Young’s modulus of the scaffolds increased with an increase of CS content, indicating that CS exhibits a better strengthening effect on the modulus when the compression direction was parallel to load. The compressive strength was shown in Figure 4b. Similar to the variation trend of Young’s modulus, the compressive strength of scaffolds was enhanced with the increase of the raster angle. When under the same printing raster angle, the strength first increased and then decreased along with the CS content ranging from 0 to 40%, demonstrating that the increase in porosity also reduced the compressive strength. The adjustable range of the compressive strength was similar to the variation range of the natural trabecular bone (2–17 MPa) [39,40]. Thus, the printed scaffolds exhibited potential for clinical application.

The compressive process of the pure PEEK scaffolds was recorded by a high-speed camera, as shown in Figure 5. There was a larger angle between the struts of scaffold and compressive loading when the raster angle was less than 45°. The struts could not bear the compressive load directly and the gap between adjacent struts gradually decreased. The scaffolds were finally compressed into nearly compacted solids when the strain reached up to 40%. The above deformation process resulted in a lower compressive modulus and strength, as shown in Figure 5 When the raster angle was more than 45°, the struts of the scaffold could bear the compressive load because of the lower angle between the struts and compressive load. Obvious buckling deformation of struts for the PEEK-60° when the strain reached up to 20% could be found. When the compressive load was further applied, a crush band was formed because of the accumulation of buckling deformation. Buckling deformation of the struts was found for the PEEK-75° and PEEK-90° when the strain reached 10%, indicating that a larger raster angle could reduce the strain corresponding to the beginning of buckling deformation. Benefiting from the excellent toughness, there was mainly plastic deformation for the pure PEEK scaffolds during the compressive process, and no obvious fracture was found.

Figure 6 and Figure 7 exhibit the compressive process of the PEEK/CS composite scaffolds. For the 20%CS scaffolds, the compressive process was similar to that of pure PEEK, indicating that the addition of CS has little effect on the deformation process when the CS content was less than 20%. For the 40%CS scaffolds, the 40%CS-15° and 40%CS-30°scaffolds were compressed into compacted solids when the strain reached up to 40%. A crack was found for the 40%CS-45° scaffold as the strain reached up 40% (shown by the red dotted lines). With the increase of the raster angle, the strain corresponding to the appearance of the crack decreased. The brittle fracture of 40%CS-75° occurred when the strain reached 10%. Thus, there was plastic deformation for 40%CS scaffolds as the raster angle was less than 45°. The plastic deformation was shifted to brittle deformation when the raster was more than 45°. The bioactivity of PEEK/bioceramic composites was usually positively associated with bioceramics content, while the participation of bioceramic may lead the decrease of toughness. Although the PEEK/CS composites with 40% CS content exhibited high brittleness, the deformation process and mechanical properties could be adjusted by changing the raster angle in this study. According to the above results, the lower raster angle could be selected to improve the load transmitting mode and avoid the brittle fracture of PEEK/CS composites with a higher bioceramic content.

Figure 8 exhibits the impact of the CS content and raster angle on the modulus and strength of the scaffolds. It could be found that the modulus and strength was enhanced with increased raster angle. The modulus and strength first increased and then decreased with the CS content ranging from 0 to 40%. The mapping relationships among the modulus, strength, raster angle and CS content were further derived by polynomial fitting, shown as Formulas (2) and (3). In clinical application, the appropriate raster angle and CS content of the scaffolds should be selected and designed according to the requirements of the mechanical properties. The mapping relationship could be used in controlling the mechanical properties of scaffolds to match that of native bone. The better loading transmittance could be realized by balancing the mechanical mismatch between the PEEK/CS scaffolds and the natural bone.
(2)$$\sigma =-2.586+0.413x+0.311y-0.011x^{2}-0.001y^{2}+0.0001xy\;\left(R^{2}=0.90616\right)$$
(3)$$E=-59.42+4.453x+5.53y-0.138x^{2}+-0.0003y^{2}+0.034xy\;\left(R^{2}=0.96389\right)$$
where *E* and *σ* are the Young’s modulus and strength under the compressive load; x is the CS content and y is the raster angle.

Some limitations of this work must also be pointed out. The mechanical behaviors of the scaffolds were studied under static loading condition. The fatigue behavior was also crucial for the clinical application of scaffolds. We found that the addition of CS particles reduced the toughness. Meanwhile, the participation of CS particles resulted in a rougher morphology and brittle fracture. The stress concentration was more likely to occur on rough surfaces of PEEK/CS scaffolds under compressive load, which further lead to the propagation of cracks and brittle fracture. Furthermore, CS was a degradable material and the degradation process of CS particle had a significant impact on the mechanical properties. Thus, the fatigue behaviors and degradation process of the PEEK/CS scaffolds should be further investigated.

## 4. Conclusions

In this work, the PEEK/CS composites scaffolds with different CS contents and raster angles were printed by fused filament fabrication (FFF) successfully. The printed PEEK/CS composite scaffolds exhibited good interconnectivity. The crystallinity degree and surface roughness of PEEK/CS scaffolds was enhanced with the CS content ranging from 0 to 40% wt. The compressive modulus of the PEEK/CS scaffolds could be tuned in the range of 23.3–541.5 MPa, similar to the variation range of natural cancellous bone. The deformation process and failure modes could be adjusted by changing the raster angle, which could avoid the brittle fracture caused by excessive bioceramic. The mapping relationship between the mechanical properties, raster angle and CS content was established, giving guidance on the selection of printing parameters and the control of mechanical properties.

## Figures and Tables

**Figure 1 polymers-13-02547-f001:**
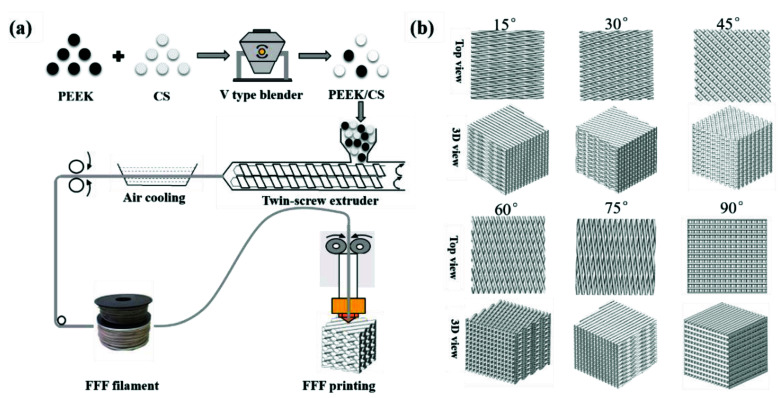
(**a**) PEEK/CS scaffolds fabricating process; (**b**) the schematic diagram of the scaffolds with different raster angle.

**Figure 2 polymers-13-02547-f002:**
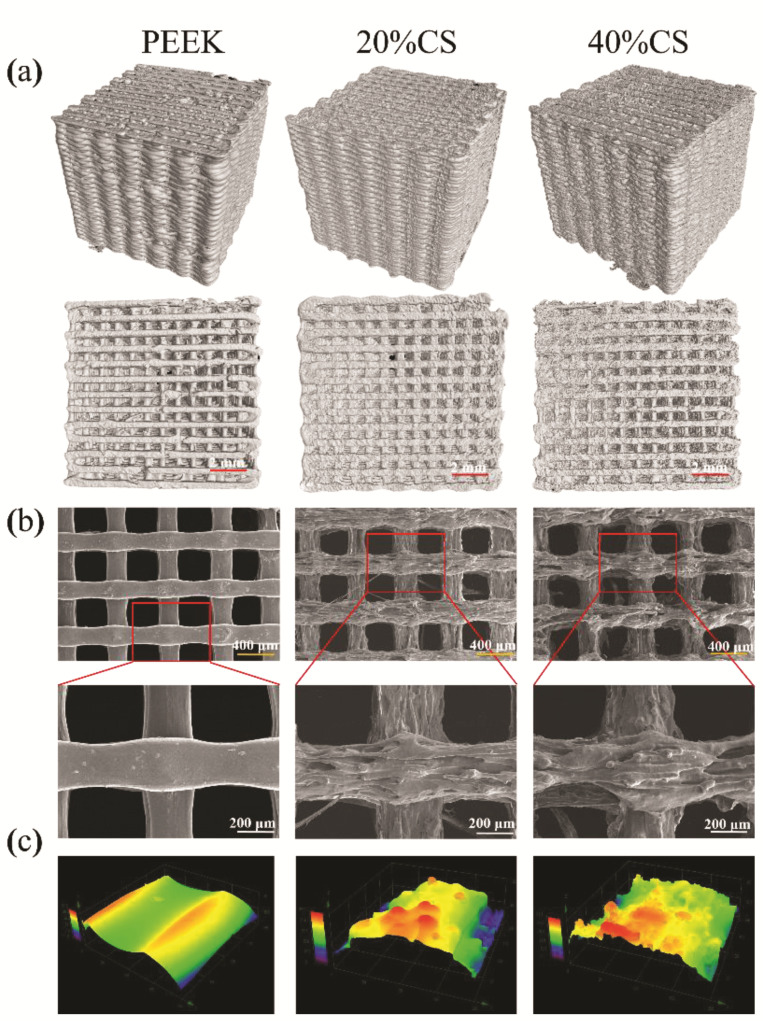
Micro-structure characteristics of the PEEK/CS scaffold samples: (**a**) geometry observation by micro-CT, (**b**) surface morphology by SEM and (**c**) three-dimensional topography by LSCM.

**Figure 3 polymers-13-02547-f003:**
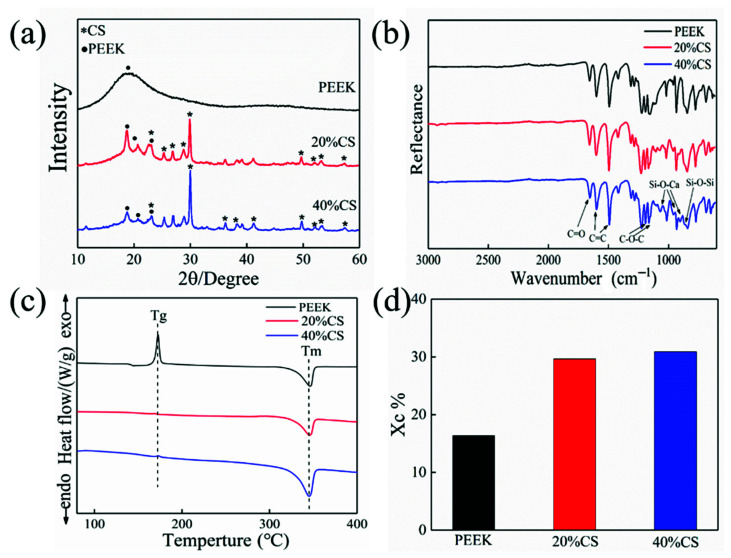
(**a**) XRD patterns, (**b**) FTIR spectra, (**c**) DSC curves, (**d**) crystallinity results of PEEK and PEEK/CS composites.

**Figure 4 polymers-13-02547-f004:**
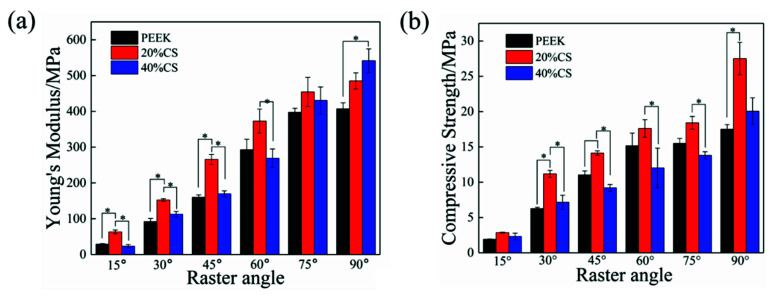
The modulus (**a**) and strength (**b**) results of the scaffolds with different CS content and raster angles.

**Figure 5 polymers-13-02547-f005:**
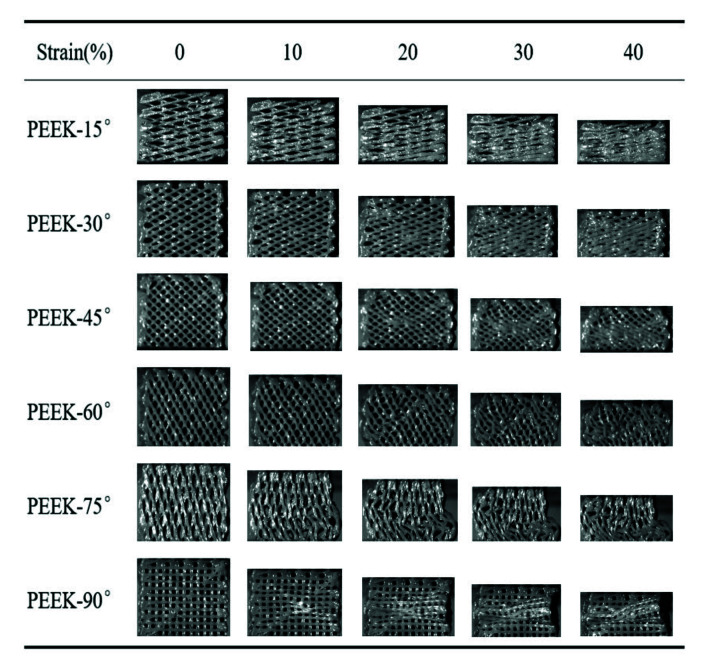
The compressive process of the pure PEEK scaffolds with different raster angles.

**Figure 6 polymers-13-02547-f006:**
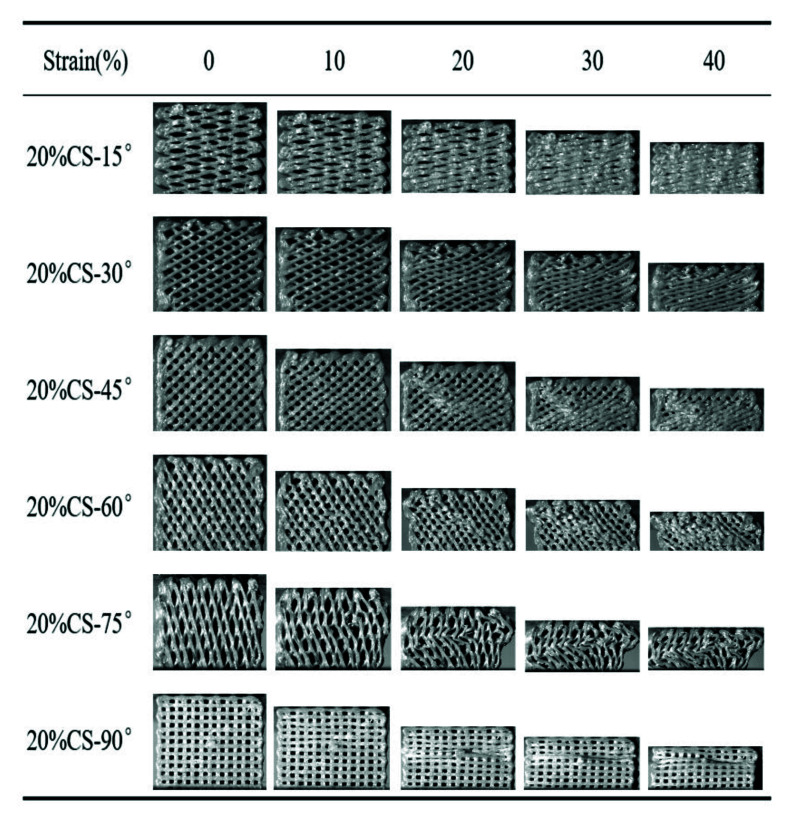
The compressive process of the 20%CS scaffolds with different raster angles.

**Figure 7 polymers-13-02547-f007:**
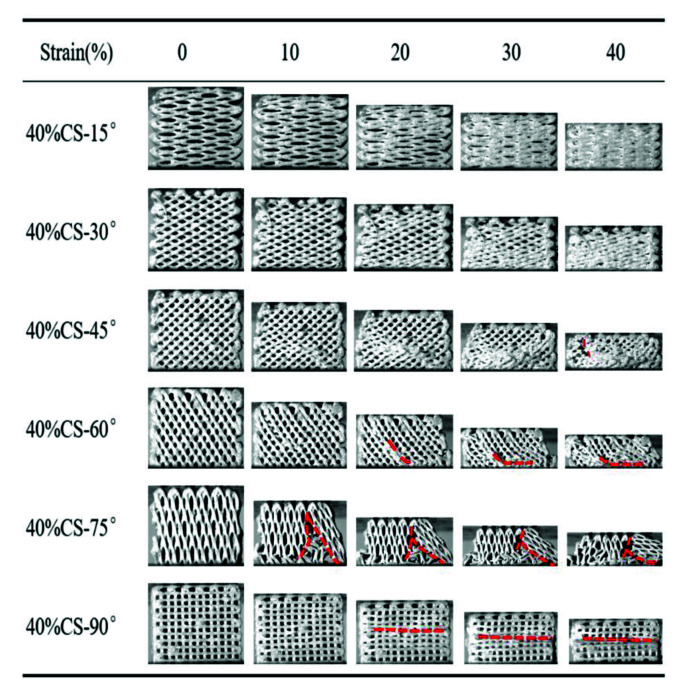
The compressive process of the 40%CS scaffolds with different raster angles.

**Figure 8 polymers-13-02547-f008:**
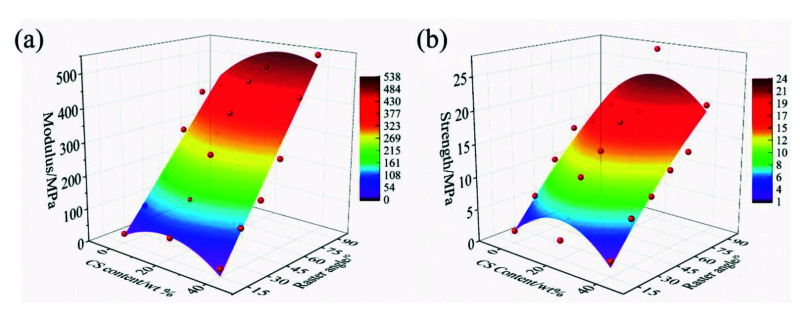
The changing law of modulus (**a**) and strength (**b**) with different CS content and raster angle.

**Table 1 polymers-13-02547-t001:** The porosity of three scaffolds with different raster angle.

Raster Angle	±15°	±30°	±45°	±60°	±75°	±90°
Pure PEEK	55.8 ± 0.76%	55.2 ± 0.38%	55.1 ± 0.61%	55.1 ± 0.55%	54.9 ± 0.46%	53.2 ± 0.20%
20% CS	55.2 ± 0.93%	54.2 ± 0.74%	54.6 ± 1.20%	54.1 ± 0.76%	55.5 ± 0.3%	56.3 ± 1.91%
40% CS	62.1 ± 0.57%	62.4 ± 0.52%	62.9 ± 0.43%	62.2 ± 0.31%	62.0 ± 0.64%	61.7 ± 0.56%

## Data Availability

The data presented in this study are available on request from the corresponding authors.

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
