# Peer review of "Effects of Raster Angle and Material Components on Mechanical Properties of Polyether-Ether-Ketone/Calcium Silicate Scaffolds"

_polymers, 2021, doi:10.3390/polym13152547_

Round 1
Reviewer 1 Report
In such work, the authors investigate the effects of 3D-printing raster angle and bio-compatible material mixtures based on homemade filaments; such pieces are emulating pieces of bone in order to be proposed as surgical implants that might replace the conventional metal-based materials. In particular they fabricate and characterize pieces based on polyether-ether-ketone that are bio-compatibility activated with calcium silicate. Although such materials have been proposed separately under other fabrication schemes, they investigate the possibility of achieving the optimum mixture that with together with the right fabrication techniques may fit the termo-mechanical properties of human bones. They demonstrated that the printed scaffolds exhibited potential for clinical applications.
In my opinion, the work is correctly exposed and the experiments, methodology and results are sufficient to add up to knowledge in the field.
Please consider the following minor points as suggestions to possibly improve your manuscript.
Line 83: in “Figure 1(a) exhibited the fabrication process of the scaffolds based FFF 3D printing.” Please consider using “exhibits” instead of “exhibited”. Please correct the same for lines 89,138,156,192,222 and 242. “Shows” instead of “showed”.
In line 98 please consider to change “2.2. Characteristics of the composite scaffolds” for “Methodology for the characterizations of the composite scaffolds”
In line 105 please use “on” instead of “by”: The porosity of scaffolds with different CS content was further calculated based by weighting method.
Good Job!
Reviewer 2 Report
The authors presented a manuscript dealing with a hot topic, such as that of Additive manufacturing and the optimization of materials composition and printing parameters. The Introduction probably needs a slight revision, including specific references for strengthening some points (see attached .pdf), but in general the paper is well structured. In my opinion, the authors must revise the use of decimal places in standard deviation and definitively improve the thermal aspect, both in the experimental than in the discussion. DSC figure needs an indication of the dimensional unit and overall exo/endo versus. The attribution of Tg is questionable (see attached .pdf) and the crystallinity needs more discussion. Finally, a strong revision of the text is recommendable, considering many typos highlighted in the attached report.

Round 2
Reviewer 2 Report
The Authors carefully revised their manuscript in agreement with the reviewers' suggestions, in my opinion the work can be published on Polymers.